# Impact of High Concentrations of Cellulose Fibers on the Morphology, Durability and Protective Properties of Wood Paint

**Massimo Calovi *** and **Stefano Rossi**

Department of Industrial Engineering, University of Trento, Via Sommarive 9, 38123 Trento, Italy
* Correspondence: massimo.calovi@unitn.it; Tel.: +39-0461-282442

**Abstract:** This work aims to reveal the effect of a high amount of cellulose fibers on the durability and protective behavior of a bio-based wood paint. The influence of the filler on the morphology of the coatings was investigated by scanning electron microscopy observations, while the durability of the paint was evaluated by exposing the samples to UV-B radiation and continuous thermal shocks. Infrared spectroscopy analysis, colorimetric inspections, adhesion tests and scanning electron microscope observations were employed to assess the role of the high concentrations of fillers in affecting the resilience of the acrylic matrix. Moreover, the impact of the filler on the barrier performance of the coatings was estimated using a liquid resistance test and a water uptake test. Finally, the mechanical properties of hardness and abrasion resistance of the layers were evaluated by means of the Buchholz Hardness Indentation test and the Scrub test. Ultimately, this study demonstrates the pros and cons of using large amounts of cellulose fibers as filler in wood paint: the work warns against the excessive use of these fibers, which need a threshold limit so as not to significantly change the coating's structure and thereby weaken its protective properties.

**Keywords:** cellulose fibers; bio-based filler; paint durability; coating protective behavior; wood paint





## 1. Introduction

Due to a variety of interesting chemical and physical characteristics [1], as well as the fact that it is a plentiful and simple element, wood is a material that has been frequently employed by humans since antiquity [2]. The consumption of wood has increased throughout time [3] thanks to its high strength-to-weight ratio [4], simplicity of processing [5], and carbon neutrality [6]. Moreover, wood is recognized for its distinct aesthetic qualities [7]. Thus, today, wood is employed in several industrial fields, such as in construction [8], architecture [9], the furniture market [10], and interior design [11]. Yet, the lignocellulose characteristics of this material, such as its high flammability [12] or low durability when subjected to humidity [13] and solar radiation [14], has frequently decreased, limiting the use of wood in several industrial domains.

Hence, whether they are used for indoor [15] or outdoor [16] applications, wooden products are frequently coated with organic coatings, with the aim of withstanding these decay events and enhancing their lifetime. These varieties of coatings primarily protect the hardwood material against UV rays [17], humidity variations [18], chemical agents [19], mechanical damage [20], and the spread of pathogenic organisms such as fungi [21].

The widespread usage of wood in exterior applications has compelled academia and business to search for novel approaches to improve the various performances of wood coatings [22]. For instance, some studies revealed that the addition of various nanoparticles, such as $TiO_2$ [23], ZnO [24], $SiO_2$ [25], and $CeO_2$ [26], has enhanced the UV light resistance of wood coatings. On the other hand, nanoparticles with high hardness, stiffness, and thermal stability, such as nanosilica [27], nanoalumina [28], nanoclay [29], and nanocellulose [30], can be used to increase the mechanical properties and water repellency

of wood coatings [31]. Moreover, with the application of nanomaterials such as copper nanopowders [32], nanotitanium [33], and silver [34,35], wood paint can gain significant antibacterial properties. Finally, using copper [36], silver nanoparticles [37], or combining fungicidal agents such as propiconazole and thiabendazole [38], the degradation of wood caused by fungal activity has been significantly reduced.

The circular economy and bio-based resources, two of the most popular topics right now, are also attracting the attention of the wood coating business. The irresponsible linear economy of the previous century triggered an irreversible rise in raw material prices and depletion of resources, which resulted in trash accumulation and irreparable ecological damage [39]. Today, a new sustainable economy is being developed as a partial answer to these issues, based on essential components such the circular production/consumption system, with the purpose of reducing the ecological footprint of the manufacturing applications [40]. The new strategy aims to maximize waste disposal systems by substituting petroleum-derived materials with renewable feedstock, including bio-based resources. While the coatings industry is increasingly focused on ecological and multifunctional alternatives to the standard synthetic fillers [41], whose production typically does not consider the aspects of environmental sustainability [42], the scientific interest in employing natural additives is constantly growing [43].

From this perspective, cellulose represents a true bio-based resource to be exploited to increase the performance of wood coatings. Several works used this material in the form of nanocrystals in order to improve the mechanical properties of organic coatings, such as the Young's modulus [44,45], hardness and abrasion resistance [46,47]. Moreover, some studies have observed an improvement in the thermal stability of the coating [48], but also an unsatisfactory bactericidal performance [49]. At the same time, cellulose nanofibers exhibited interesting improvements in the mechanical properties of organic coatings [20,50–52], as well as UV and oxygen shielding phenomena [53]. However, the use of this filler is always limited to low concentrations to avoid undesired phenomena of voids and porosity in the bulk of the coating [54], or a modification of the rheology of the paint with a consequent sudden increase in viscosity [55]. Furthermore, the production of nano-sized fillers does not comply with the aspects of sustainable economy and the large-scale use of bio-based products.

Therefore, this work aims to investigate the possibility of using large quantities of micro-scale cellulose-based fibers in wood coatings, the production of which respects the principles of the circular economy. In fact, the cellulose fibers supplied by Rettenmaier Italia (Castenedolo, Italy) are obtained from wood pulp, employing renewable materials and wood waste, by means of basic mechanical processes such as grinding, fractionation and air classification (grain size adjustment). As the actual circular production/consumption system favors the increasingly massive use of renewable resources, the current application of limited quantities of fibers as filler in coatings does not produce a real impact on the economy of the sector. Therefore, this study employs high and unusual quantities of fibers compared to previous literature works in order to reveal the pros and cons of mass use of cellulose-based fillers in the paint field.

Three different amounts of cellulose fibers were introduced into a commercial acrylic bio-based wood paint to evaluate the effect of a high concentration of filler on different features of the coating. The morphology of the layers was investigated with scanning electron microscopy (SEM) observations, while the durability of the coatings was estimated by means of a climatic chamber and UV-B exposure. The eventual decay of the samples due to these two accelerated degradation tests was monitored with colorimetric analyses, scanning electron microscope observations, a Cross Cut Test and infrared spectroscopy measurements. The influence of the cellulose fibers on the barrier properties of the paint was studied with the chemical resistance test and liquid water uptake test, rating possible changes on the appearance of the samples. Moreover, the mechanical properties of the coatings and the role of the fibers were evaluated by means of the Buchholz Hardness Indentation test and the Scrub test.

## 2. Materials and Methods

### 2.1. Materials

The cellulose fibers ARBOCEL BE 600/30 PU were supplied by Rettenmaier Italia (Castenedolo, Italy) and used as received. The $150 \times 150 \times 2$ mm$^3$ poplar wood substrates were supplied by Cimadom Legnami (Lavis, Italy). The acrylic bio-based paint TECH20 was provided by ICA Group (Civitanova Marche, Italy). It is formulated with raw materials from renewable sources. Sodium chloride ($\geq$99.0%) and ethanol (99.8%) were purchased from Sigma-Aldrich (St. Louis, MO, USA) and used as received. The commercial detergent disinfectant product Suma Bac D10 Cleaner and Sanitiser (Diversey—Fort Mill, SC, USA), containing benzalkonium chloride (3.0–10.0 wt.%), and the cataphoretic red ink Catafor 502XC (Arsonsisi, Milan, Italy) were purchased and used for the liquid resistance tests.

### 2.2. Samples Production

First, the poplar wood panels were grinded with 320 grit sandpaper to ensure a smooth surface. Subsequently, the bio-based paint provided by ICA Group was sprayed onto the pre-treated wooden surfaces, and the coatings were allowed to air dry at room temperature for 4 h. The entire process was repeated two times. Samples F5, F10 and F20 were realized, introducing 5 wt.%, 10 wt.% and 20 wt.% of fibers into the paint, respectively. Before the application, the three paint formulations were mechanically stirred for 30 min. The behavior of the three composite coatings was compared with the performance of sample F0, free of fibers, used as a reference. The four sample series are summarized in Table 1 with the samples nomenclature.

**Table 1.** Samples nomenclature.

| Samples Nomenclature | Fibers Amount in the Paint Formulation (wt.%) |
|----------------------|-----------------------------------------------|
| F0                   | 0                                             |
| F5                   | 5                                             |
| F10                  | 10                                            |
| F20                  | 20                                            |

### 2.3. Characterization

The low-vacuum scanning electron microscope SEM JEOL IT 300 (JEOL, Akishima, Tokyo, Japan) was used to observe the fibers' appearance and to investigate the surface and the cross section of the coatings, with the aim of evaluating the effect of the high amount of fibers on the layers' compactness and structural morphology.

The samples were subjected to two accelerated degradation tests, simulating exposures in aggressive environments, to assess the possible role of the cellulose fibers in modifying the durability of the paint.

The climatic chamber ACS DM340 (Angelantoni Test Technologies, Perugia, Italy), was employed to simulate extreme thermal changes. According to the UNI 9429 standard [56], the exposure test consisted of 15 cycles of:

- Four hours at +50 °C and relative humidity <30%;
- Four hours at −20 °C;
- Sixteen hours at ambient temperature.

To limit the moisture absorption by the wood substrate, the 5 uncoated surfaces of the $40 \times 40 \times 2$ mm$^3$ poplar wood sample were completely sealed with silicone. The samples were monitored with colorimetric analyses every 3 cycles of exposure in the climatic chamber to investigate the effect of the fiber concentration in altering the paint durability. The colorimetric analyses were performed with a Konica Minolta CM-2600d spectrophotometer (Konica Minolta, Tokyo, Japan) with a D65/10° illuminant/observer configuration in SCI mode. Moreover, the changes in the adhesion of the coatings due to the thermal cycles were evaluated with the Cross Cut Test, following the ASTM D3359-17 standard [57], performed after the exposure in the climatic chamber. According to the

standard, the samples were carved with the appropriate cutting tool to form a grid of $6 \times 6$ chess pieces, each measuring 1000 μm $\times$ 1000 μm. The standardized tape was applied to the grid and removed after 30 s, with an orientation of 45° with respect to the sample surface. Thus, the samples were analyzed with an optical microscope Nikon SMZ25 (Nikon, Tokyo, Japan), to observe possible removal of the coating from the cuts.

The resistance of the coatings to UV light was characterized, exposing the samples in a UV-B chamber UV173 Box Co.Fo.Me.Gra (Co.Fo.Me.Gra, Milan, Italy) for 200 h, according to the ASTM G154-16 standard [58]. The possible decay of the layers was assessed with FTIR infrared spectroscopy measurements and colorimetric analyses. The FTIR spectra were acquired with a Varian 4100 FTIR Excalibur spectrometer (Varian, Santa Clara, CA, USA), to assess the chemical modifications of the polymeric matrix.

The effect of different fibers amount on the barrier performance of the acrylic matrix was studied with the chemical resistance tests, according to the GB/T 1733-93 standard [59]. The layers behavior was estimated by dipping a filter paper in 15% sodium chloride solution, 70% ethanol, detergent, and red ink, respectively. Subsequently, the filter paper was placed on the surface of the coatings and covered with a glass cover. After 24 h, the glass cover and filter paper were removed and the residual liquid on the coating surface was absorbed. The imprint and discoloration were evaluated with the colorimetric analyses. Moreover, possible changes in coatings adhesion due to the test solution absorption were assessed with the Cross Cut Test, carried out before and after the liquid resistance test. Furthermore, the actual barrier properties of the paints against the water absorption were rated with the Liquid Water Uptake test, following the EN 927-5:2007 standard [60]. As for the climatic chamber exposure, the 5 uncoated surfaces of the $40 \times 40 \times 2$ mm$^3$ poplar wood panels were completely sealed with silicone to avoid water uptake phenomena by the wooden substrate. The samples were preconditioned at 65% RH and 20 °C, after which they were set to float in a container with water. The mass gain was measured after 0, 6, 24, 48, 72, and 96 h to determine the moisture uptake, expressed in g/m$^2$.

Finally, the influence of the cellulose fibers on the mechanical properties of the acrylic coating was investigated by means of the Buchholz Hardness Indentation test and the Scrub test. The Buchholz test was performed following the ISO 2815 standard [61], measuring the length of the indentation realized by the standardized instrument. The Scrub test was carried out, employing an Elcometer 1720 Abrasion and Washability Tester (Elcometer, Manchester, UK), following the ISO 11998 standard [62]. The coatings mass loss was monitored every 250 cycles (37 cycles per minute), for a total of 1000 cycles, to evaluate their resistance to abrasion. Unlike that which is described in the standard, the test was carried out in dry mode, without the aid of a cleaning solution, in order to avoid absorption phenomena of the test solution in the polymeric matrix and in the wood, with consequent falsification of the results.

## 3. Results and Discussion

### 3.1. Fibers and Coatings Morphology

Figure 1 reveals the aspect of the cellulose fibers, observed by SEM. The fibers were positioned on the appropriate SEM support, by gluing with conductive tape. The fibers show irregular morphology, alternating linear shapes with thicker structures, similar to granules. The latter do not exceed 50 μm in size, while the filamentous structures can reach and exceed 100 μm in length. However, the tendentially two-dimensional morphology of the fibers makes them an interesting bio-based filler for applications in coatings with thickness well below 100 μm.

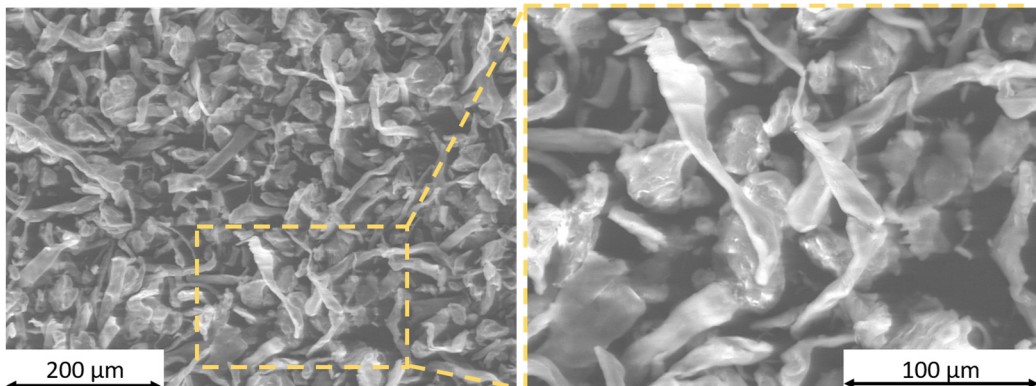

**Figure 1.** SEM micrographs of the cellulose fibers. Low magnification on the left (200×) and higher magnification on the right (500×) with the focus on the morphology of the fibers.

Thus, the cellulose fibers were introduced in the formulation of the bio-based paint, increasing the filler content from 5 wt.% to 20 wt.%. The samples were subjected to a brittle fracture process in liquid nitrogen, with the aim of using SEM to examine the compatibility of the fibers with the polymeric matrix, and to explore the effect of the high amount of fibers on the structural morphology of the coatings. Figure 2 shows the SEM micrograph of both the top view (on the left) and the cross section (on the right) of the four kinds of coatings. Figure 2a, relative to the reference sample F0, free of filler, exhibits a compact and defect-free layer, typical of the layers deposited with the spray technique [63], with thickness of about 140 μm. On the other hand, the introduction of cellulose fibers into sample F5 (Figure 2b) leads to a considerable increase in layer thickness, which reaches 250 μm. This clear growth in size is due to the increase in the amount of material in the paint formulation: since the fibers are particularly light, the volume occupied by the 5 wt.% is relevant. In fact, while the specific weight of the paint is between 1.01 and 1.18 g/cm$^3$, the bulk density of the fibers is around 0.22 g/cm$^3$. Thus, the large amount of fibers is appreciable both in the top view and, above all, in the cross section of the coating. Despite the significant presence of fillers within the bulk of the coating, the layer appears free from macroscopic defects and the fibers show good compatibility with the polymeric matrix. However, the further increase in fiber concentration causes a substantial change in the morphology of the composite layer, as evidenced in samples F10 and F20 (Figure 2c,d, respectively). The 10 wt.% of filler in sample F10 causes different agglomerations of fibers, clearly visible both in the top view and in the cross section. Consequently, the layer is not homogeneous, but shows a morphology characterized by different accumulations of material. The thickness of the coating is not constant, but varies from 100 to 300 μm. However, despite this undesirable phenomenon, the coating appears to completely coat the wooden substrate. Sample F20 also exhibits a layer that completely covers the substrate, but in this case, the coating does not seem to provide the adequate protective guarantees. It is incorrect to speak of agglomeration of filler, as the entire coating appears to be composed mainly of fibers. The amount of fibers is so high as to represent the major component of the layer. Rather, it appears that the various fibers are interconnected by a thin layer of paint. These phenomena result in a highly defective coating, with significant interconnected and open porosity. It is not possible to determine the real thickness of the layer, which is mainly composed of voids and randomly distributed fibers. A previous work in the literature [55] has in fact underlined the important issue that has to be taken into account when applying cellulose fibers as additives in water-based coating systems. Similarly, another study has suggested employing a low amount of nanofibers to avoid the development of defects and porosity in the bulk of the coating [54].

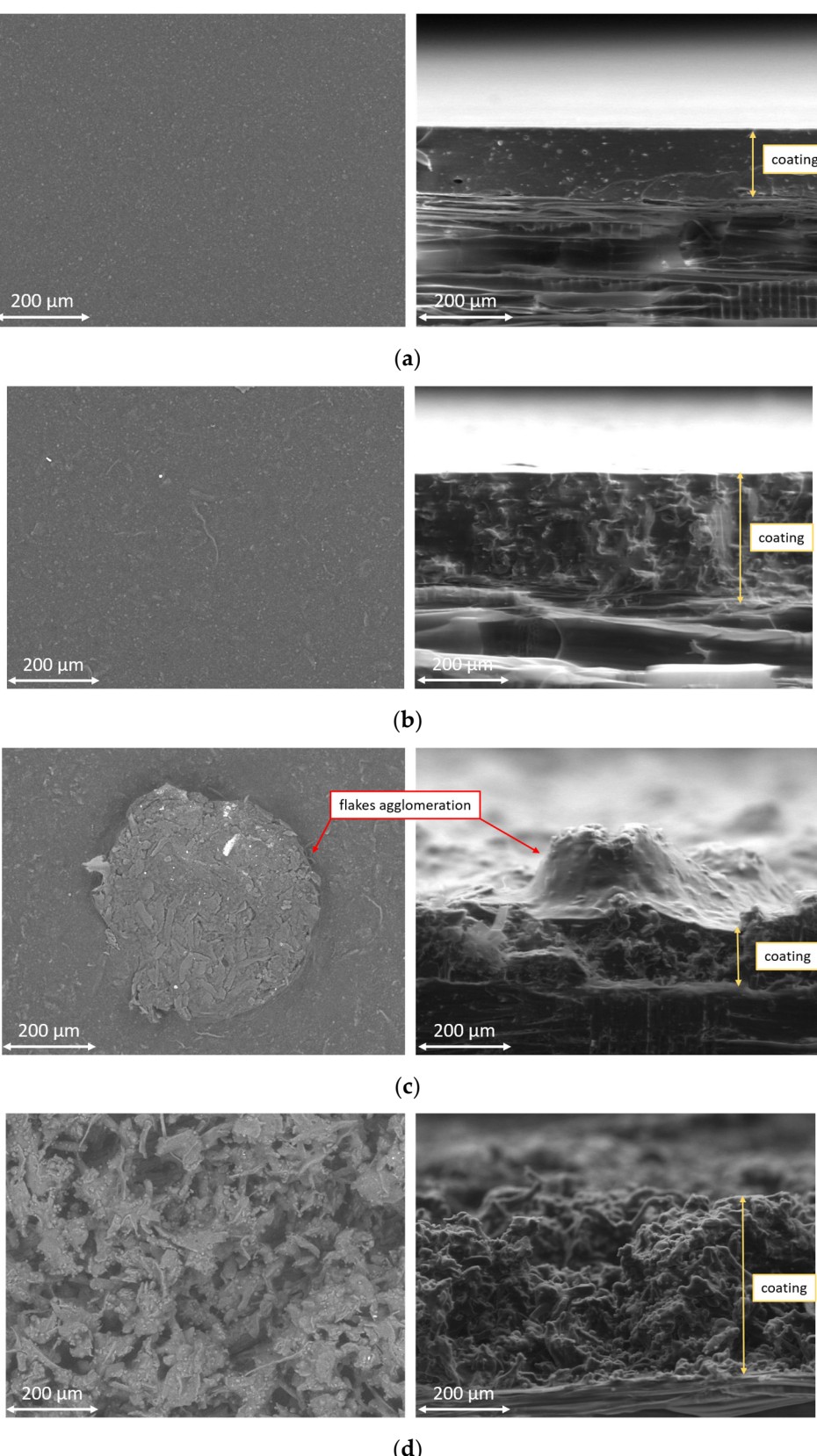

**Figure 2.** SEM micrograph of (**a**) sample F0, (**b**) sample F5, (**c**) sample F10 and (**d**) sample F20. Top view on the left and cross section on the right.

The SEM observations suggest that the protective performance of the coating is reduced as the concentration of fibers increases, as they introduce significant defects in the

polymeric matrix. Thus, it is important to take into account that using a large amount of cellulose fibers as a filler in paints results in a change in the layer's appearance.

### 3.2. Sample Exposure to Different Aggressive Environments

The surface morphology and the texture of the coating can be influenced by this type of bio-based filler: however, this aspect could represent an aesthetic multifunctionality of the fibers. Nevertheless, it is necessary to evaluate how this filler, in large quantities, affects the protective performance of the paint. Thus, the samples were subjected to accelerated degradation tests, such as exposure to UV-B radiation and extreme temperature changes, in order to assess the impact of the different amounts of fibers.

#### 3.2.1. UV-B Exposure

Figure 3 represents the FTIR spectra of the four coatings and of the wooden panel, before and after the exposure of 200 h to the UV-B radiation. The wooden substrate shows the same peaks in the two spectra. The band between 3400 and 3300 cm$^{-1}$, is representative to the stretching vibration of the –OH group [64], while the stretching region between 3000 to 2800 cm$^{-1}$ can be associated with the –CH group of cellulose, hemicellulose, and lignin [65]. The peak at 1729 cm$^{-1}$ corresponds to the stretching vibrations of the unconjugated C=O group and specific moieties of the polymeric chains present in the wood, such as esters [66]. The bands at 1592 cm$^{-1}$ and 1460 cm$^{-1}$ can be associated with the C=C benzene ring vibration of lignin and C–H deformation vibration, respectively [67]. The last two peaks at 1233 cm$^{-1}$ and 1027 cm$^{-1}$ are representative of the C–O stretching and the typical C–O–C stretching vibrations of cellulose [68], respectively. The exposure to the UV-B radiation results in an increase in the intensity of the peak at 1729 cm$^{-1}$, which is accompanied by a reduction in the signal of the peak at 1233 cm$^{-1}$. This phenomenon corresponds to the decay of the wood panel, represented by the alteration of the cellulose chemical structure [35].

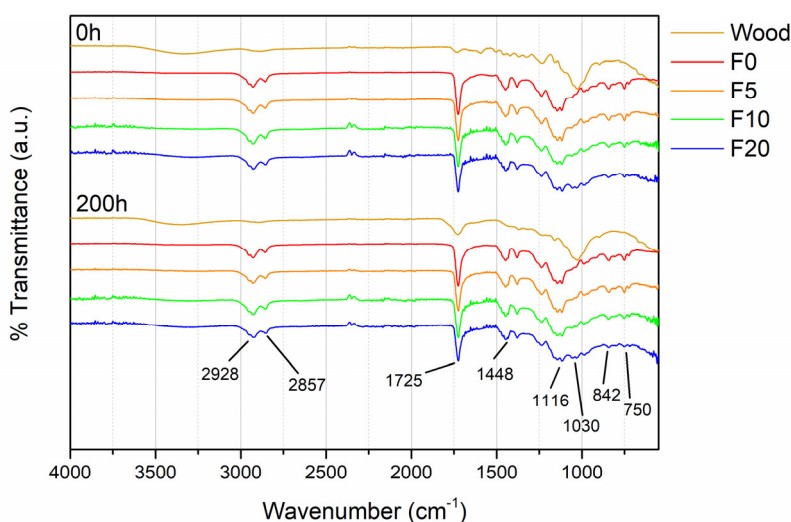

**Figure 3.** Evolution of the FTIR spectra of the samples before and after the exposure to UV-B radiation.

The FTIR spectra of the four coatings are very similar to each other. The presence of the cellulose fibers can be appreciated by the increase in intensity of the band at 1030 cm$^{-1}$, which is representative of the cellulose C–O–C stretching vibrations, as a function of the quantity of filler added to the acrylic layer.

At the same time, the peaks of the wooden substrate are completely covered by the signal of the acrylic paint. The two peaks at 2928 cm$^{-1}$ and 2857 cm$^{-1}$ refer to the CH$_3$ and CH$_2$ stretching vibrations, respectively. The intense peak at 1725 cm$^{-1}$ is attributed to the carbonyl stretching band. The signal at 1448 cm$^{-1}$ is representative of the C–H bending. The intense broad peak ranging from 1260 cm$^{-1}$ to 980 cm$^{-1}$ can be explained by the C–O

(ester band) stretching vibration. Finally, the two peaks at 842 cm$^{-1}$ and 750 cm$^{-1}$ are attributed to the C–H and C–C–C vibrations, respectively. The spectra of the four coatings do not undergo significant changes following UV-B radiation: this result confirms the excellent resistance of the acrylic paint against photooxidative degradation induced by UV light [69,70].

Despite the poor durability of cellulose to UV-B radiation, the spectrum of sample F20, containing a high amount of fibers, does not seem to be affected by exposure in the UV chamber. However, the appearance of some samples varies significantly during the UV-B radiation exposure test, as evidenced in Figure 4. The wooden panel reveals a clear yellowing, due to the decay of the chemical structure of cellulose and lignin. The cellulose and lignin constituting the poplar wood underwent degradation of their chemical structure, resulting in a clear yellowing of the surface of the wood panel. This phenomenon seems to be attenuated owing to the presence of the acrylic-based coating. However, visual observation of the samples reveals a poor protective effect of the coating containing a high concentration of cellulose fibers. The porous structure of the layer of sample F20 reveals the underlying wood, degraded by UV radiation exposure. The irregular and porous morphology of the coating, observed in Figure 2d, confirms to represent a critical aspect in relation to the protection of the wooden substrate.

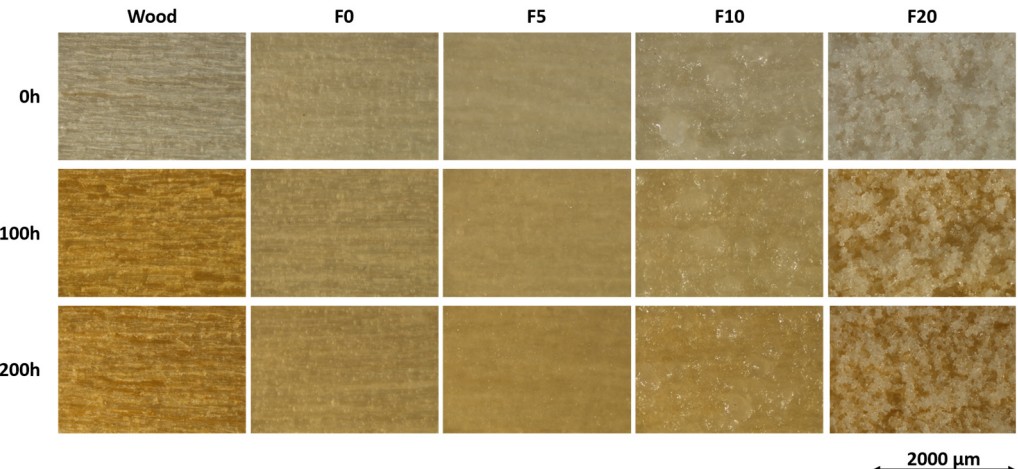

**Figure 4.** Change in the appearance of the coated samples and the wooden reference panel during exposure to UV-B radiation.

To quantify this aesthetic decay process, the color of the samples was monitored as a function of the duration of exposure to UV-B radiation. Figure 5 exhibits the evolution of the total color variation of the samples, ΔE, evaluated during the accelerated degradation test. ΔE has been calculated according to the ASTM E308-18 standard [71]:

$$\Delta E = [(\Delta L^*)^2 + (\Delta a^*)^2 + (\Delta b^*)^2]^{1/2}. \tag{1}$$

The colorimetric coordinates L*, a* and b* correspond to the lightness (0 for black and 100 for white objects), the red-green coordinate (positive values are red, negative values are green), and the yellow-blue coordinate (yellow for positive values, blue for negative values), respectively. The uncoated wooden panel shows a clear color change after only 50 h, which remains constant over time, confirming the rapid decay of its chemical–physical structure. The value of ΔE of the four coated samples increases during the exposure to UV-B radiation, a symptom of a slow but constant evolution of the state of the surfaces. As observed in Figure 4, this phenomenon is more evident with the increase in fiber content in the polymeric matrix. At the end of the test, the ΔE of the four coatings stood between values of 11 and 16. The literature defines a value of ΔE ≥ 1 as the lower limit of color change distinguishable even by the human eye [72]. Thus, the appearance of the samples is significantly affected by the exposure test to ultraviolet radiation. A recent

work [53] demonstrated that paint containing cellulose fibers shielded the UV rays more effectively than the original acryl coating, reducing the generation of radicals from the wood. However, the concentration of fibers in this work is much higher, resulting in a reduction in the compactness and an increase in the morphological instability of the layer.

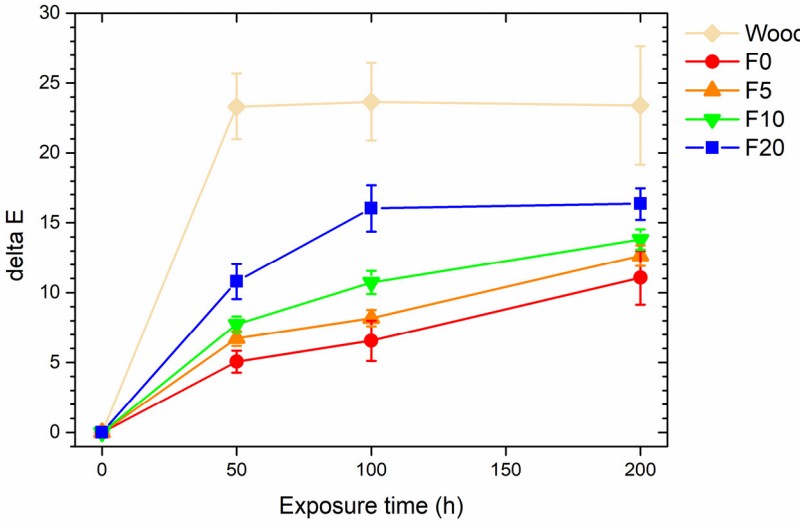

**Figure 5.** Color variation of the samples during UV-B exposure.

Ultimately, the acrylic paint revealed good chemical resistance to UV-B radiation, reducing the decay of the wooden substrate. The presence of the cellulose fibers, especially in large quantities, partially inhibits the protective role of the acrylic matrix, due to their tendency to photo-oxidize as a result of their cellulose nature and due to structural changes in the coating.

### 3.2.2. Climatic Chamber Exposure

Figure 6 reveals the evolution of the total color variation of the samples, ΔE, monitored during the exposure in the climatic chamber. The four sets of samples show an inconsistent color change, which varies during the test. ΔE is mainly due to a 1–2 point decrease in the coordinate L*, associated with an approximately 0.5–1 point increase in coordinate b*. Absolutely, the values of ΔE are not so significant as to represent a real change in the aesthetics of the coatings. Similarly, the wooden panel also does not exhibit a noticeable color change.

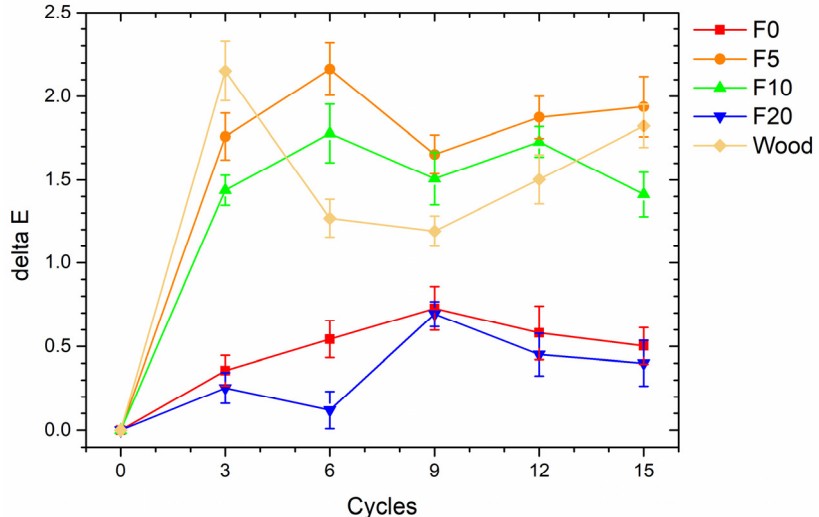

**Figure 6.** Color variation of the samples during climatic chamber exposure.

The standard [56] refers to Table 2, which divides the samples into three categories according to the level of whitening and defectiveness (cracks) developed following the climatic chamber test.

**Table 2.** Categories for whitening and cracks development following the climatic chamber exposure.

| Category | Whitening | Cracks |
|---|---|---|
| 0 | No whitening | No changes |
| 1 | Light whitening | Fractures visible only with 4× optical system |
| 2 | High whitening | Clearly visible fractures |

Therefore, according to the standard [56], the samples fall into category 0, as they do not reveal any relevant whitening phenomenon. Moreover, the samples do not manifest cracks and breakage due to the continuous changes in temperature exerted by the climatic chamber. Usually, wood paints can exhibit cracks following the exposure test in the climatic chamber [35], due to the fragile behavior of the organic coating at low temperatures and the leaching of moisture into the intrinsic porosity of the polymer matrix. However, as it was not possible to observe these defects under the optical microscope with 4× magnification, the four series of samples fall in the series 0 of the standard [56] for the 'cracks' category.

This positive result is due to the acrylic paint, which shows good durability, while the complex morphology of the coatings F10 and F20 has not allowed us to identify particular defects caused by the high concentrations of fiber. However, in order to evaluate whether the accelerated degradation test could have caused coating adhesion problems, the samples were subjected to a Cross Cut Test after the climatic chamber exposure. Figure 7 shows the Cross Cut Test results on the surface of the four series of coatings. Despite the thermal changes they have undergone, the four series of samples still highlight excellent adhesion, equal to level 5B of the ASTM D3359-17 standard [57], which is representative of no material removal following the measurement.

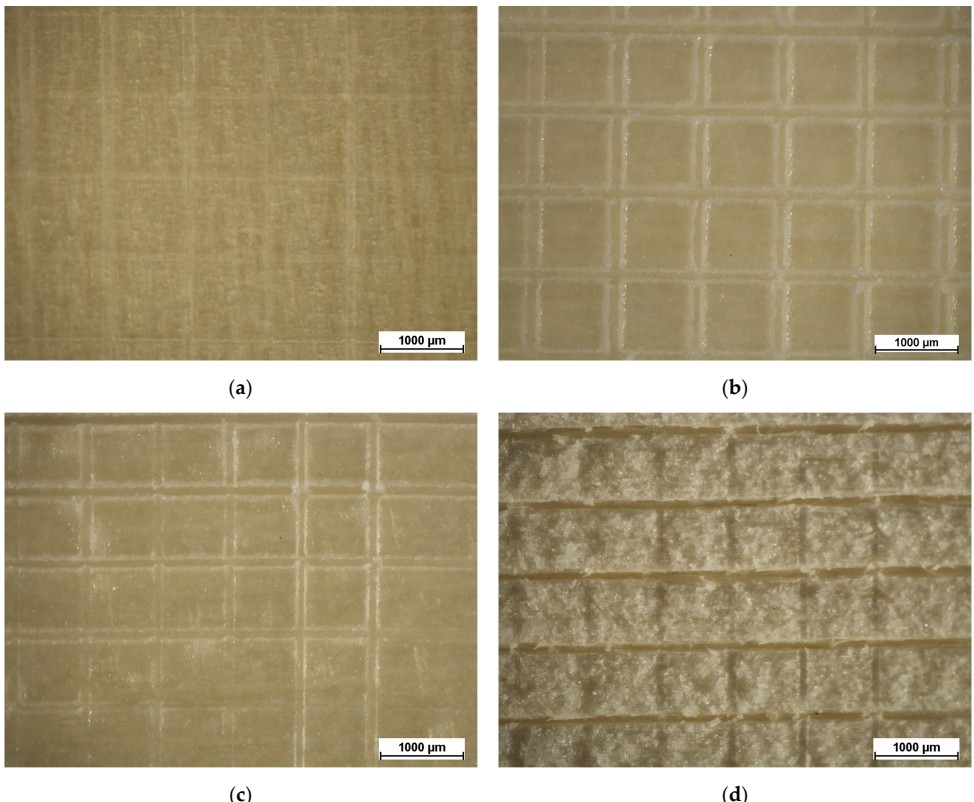

(a)      (b)

(c)      (d)

**Figure 7.** Cross Cut Test results after exposure to the climatic chamber of (**a**) sample F0, (**b**) sample F5, (**c**) sample F10 and (**d**) sample F20, observed with optical microscope.

Ultimately, all the samples reveal excellent behavior when exposed to continuous temperature changes, as the coatings do not exhibit relevant whitening phenomena or development of cracks and fractures. This output is closely connected to the adhesion of the coatings, which remains very high also following the accelerated degradation test. The cellulose fibers do not seem to affect the protective performance of the paint in environments with critical temperatures. Thus, this type of filler could also be employed for outdoor applications coatings, as long as they are not directly exposed to solar radiation, which could cause photo-oxidation phenomena in the fibers, reducing the protective feature of the paint.

### 3.3. Coatings Liquid Resistance

The effect of functional filler on the barrier properties of wood coatings is usually investigated by means of the liquid resistance test [73,74]. Table 3 correlates the values of the measured color change $\Delta E$ with the degree of discoloration shown by the coatings in contact with specific test solutions [75].

**Table 3.** Color change values corresponding to the level of discoloration.

| Level | Degree of Discoloration | Color Difference |
|:---:|:---:|:---:|
| 0 | No color change | $\leq 1.5$ |
| 1 | Very slight discoloration | 1.6–3.0 |
| 2 | Slight color change | 3.1–6.0 |
| 3 | Apparent discoloration | 6.1–9.0 |
| 4 | Severe color change | 9.1–12.0 |
| 5 | Complete discoloration | >12.0 |

Figure 8 represents the results of the chemical resistance test carried out on the four samples, including their levels of discoloration. The coatings show a negligible color change in contact with NaCl, ethanol and detergent solution, as they exhibit degrees of discoloration equal to zero. The presence of cellulose fibers does not significantly alter the behavior of the acrylic paint, even at high concentrations. Otherwise, the red ink leaves a clear halo on the surface of all four samples, whose $\Delta E$ varies by at least 14 points. Consequently, referring to Table 3, the level of discoloration is equal to 5, the highest, regardless of the type of sample. The color variation seems to be accentuated by the presence of the fibers, with a progressive increase in $\Delta E$ from sample F5 to coating F10. The cellulose fibers, whose remarkable quantity within the bulk of the coating is well highlighted in Figure 2, represent probable sites of discontinuity in the polymeric matrix, with a consequent boost of absorption of the test solution.

However, sample F20 shows similar values to the pure acrylic matrix (F0). This unexpected behavior is clarified by the images in Figure 9, which are representative of the surface of the samples after the contact with the red ink, acquired with the optical microscope. The aspect of the layer tends to become more and more red passing from the sample F0, to F5 and then F10. However, the appearance of sample F20 (Figure 9d) is different. The layer, largely made up of cellulose fibers, remains white, as the filler does not absorb the ink. Rather, the latter definitely penetrates the porous morphology of the coating until it reaches the wooden substrate, which is easily red colored. The colorimetric measurements offer a result which is partially masked by the light appearance of the cellulose fibers. The fact that the $\Delta E$ value is similar for samples F0 and F20 does not mean that their coatings offer the same protective performance. The images highlight that the layer of sample F20 does not possess good barrier properties, despite the liquid resistance test showing results in line with those of the pure acrylic matrix.

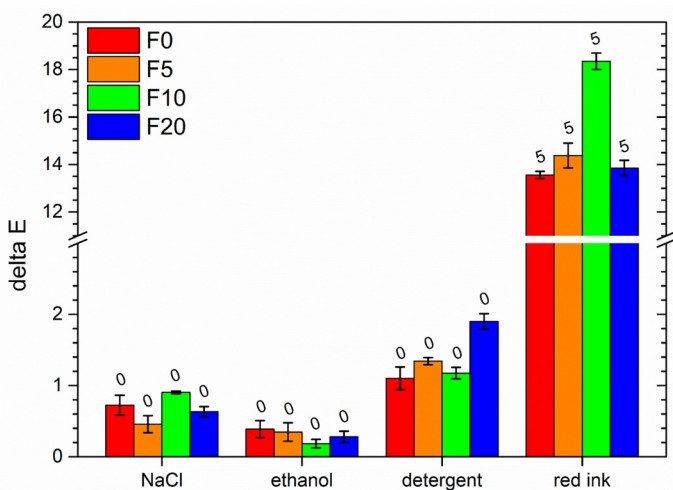

**Figure 8.** Coatings color changes of the samples after contact with liquids. The numbers above the columns are representative of the discoloration levels reported in Table 3.

|         |         |
|---------|---------|
| (**a**) | (**b**) |
| (**c**) | (**d**) |

**Figure 9.** Appearance of (**a**) sample F0, (**b**) sample F5, (**c**) sample F10 and (**d**) sample F20.

Obviously, this liquid absorption is associated with a subsequent reduction in the adhesion of the coating. Table 4 shows the results of the Cross Cut Test, carried out before and after the chemical resistance test, according to the ASTM D3359-17 [57]. Prior to contact

with the test solutions, the four coatings exhibited level 5B adhesion, representative of no material removal following the measurement. The same result was obtained after contact with NaCl solution, ethanol and detergent. The absorption of red ink due to the presence of the cellulose fibers resulted in a decrease in the adhesion degree of the coating F5, F10 and F20 to levels 4B, 3B and 2B, respectively. The material removal range associated with these three levels is less than 5%, 5% to 15% and 15% to 35%, respectively.

**Table 4.** Cross Cut Test results before and after the liquid resistance test.

| Sample | Adhesion Level | | | | |
|---|---|---|---|---|---|
| | Before the Test | NaCl | Ethanol | Detergent | Ink |
| F0 | 5B | 5B | 5B | 5B | 5B |
| F5 | 5B | 5B | 5B | 5B | 4B |
| F10 | 5B | 5B | 5B | 5B | 3B |
| F20 | 5B | 5B | 5B | 5B | 2B |

As the liquid resistance test does not issue quantitative statistics on the real barrier effect of the coatings, the samples were also subjected to the Liquid Water Uptake test. Figure 10 reveals the results of the evolution of the water uptake phenomena measured during the test. The acrylic matrix of the coatings possesses an intrinsic porosity, which leads to a continuous increase in water uptake during the test for all four samples. The slope of the curves decreases over time, as a symptom of a partial saturation of the coatings and possible absorption of the wooden substrate. The graph highlights the role of the cellulose fibers in favoring the absorption of water inside the polymeric matrix, confirming the previous results of the liquid resistance test. Despite the low absolute values of water uptake, as other literature works manifest water uptake values that can even reach 2000 g/m$^2$ after 100 h [76], the high amount of cellulose fibers compromises the protective performance of the acrylic paint. At the end of the test, compared to the pure acrylic matrix, samples F5, F10 and F20 show an increase in water uptake of 25.4%, 39.1% and 53.5%, respectively. However, the comparison of the behavior of the four samples with the output of the uncoated wood panel reveals an insulating behavior of the composite layers. The wooden panel absorbs high quantities of water, especially during the first few hours of testing. The water uptake level of the four samples is always significantly lower, suggesting an effective barrier contribution of the paint. Although this phenomenon is reduced by the presence of large quantities of cellulose fibers, at the end of the test, it is still possible to clearly differentiate the behavior of the four coatings compared to the pure wooden panel.

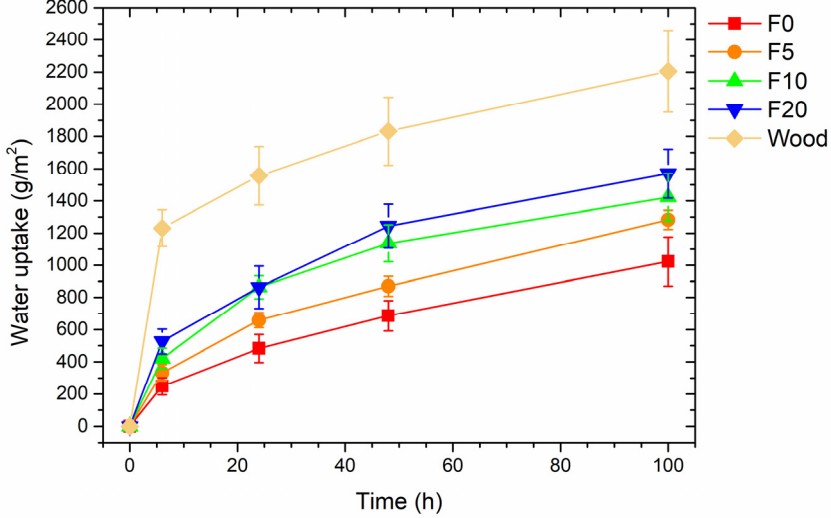

**Figure 10.** Evolution of the water uptake during the test.

In conclusion, it can be stated that the cellulose fibers can significantly influence the barrier performance of the matrix if employed at a high concentration. The filler introduces discontinuity in the polymeric matrix, favoring the uptake of liquids: this phenomenon is accentuated by a substantial change in the layer bulk structure. Thus, the results suggest limiting the amount of fibers, without exceeding the 5 wt.% employed in sample F5, to avoid unpleasant structural and rheological modifications of the paint.

### 3.4. Coatings Hardness and Abrasion Resistance

The graph in Figure 11 shows the average length of the indentations (10 measurements per sample) produced by the Buchholz disc indenting tool, with the corresponding Buchholz hardness value. On several occasions, cellulose fibers have exhibited good hardness and the ability to increase the mechanical properties of polymeric matrix composites [77–79]. Limited quantities of fibers (sample F5) cause an effective improvement of the hardness of the coating. The cellulose fibers, which are well distributed within the polymeric matrix, provide it with a better mechanical performance. A gradual increase in fiber concentration, on the other hand, results in a constant decrease in coating hardness. Although its average measured value is lower than that of the pure acrylic matrix, sample F10 exhibits large standard deviation. This phenomenon resides in the non-homogeneous surface morphology of the coating, which presents various asperities shown in Figure 2c. The average hardness value is the result of a 'hard' contribution resulting from the agglomerations of fibers and a 'soft' component provided by the acrylic matrix. Otherwise, the 20 wt.% of fibers cause a substantial decrease in the hardness of the coating. This outcome is due to the structure of the coating, which is highly porous and not very compact, and thus easily yields to the action of the Buchholz indenter. Additionally, in this case, the standard deviation of the mean value is particularly significant, as the non-homogeneity of the coating produces non-constant hardness values.

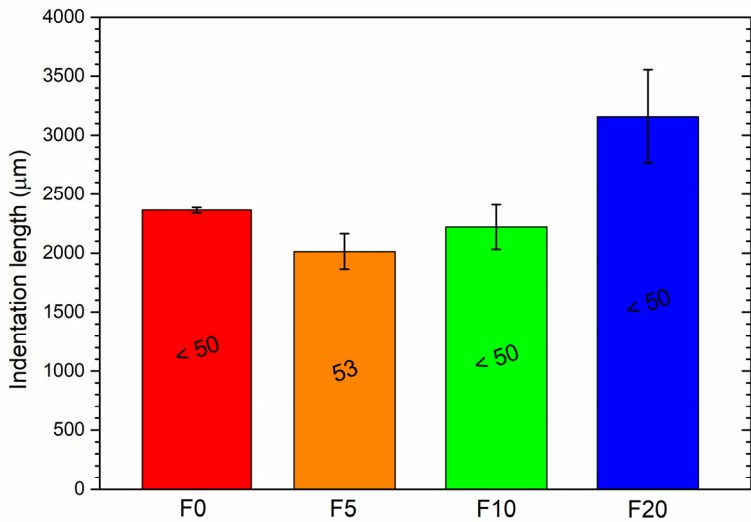

**Figure 11.** Evolution of the indentation length of the Buchholz test notches, with the corresponding Buchholz hardness values.

Some literature works describe the increase in the abrasion resistance of paints by adding cellulose nanofibers [20,52] and nanocrystals [47]. However, the size and quantity of the filler is always very limited. Therefore, this work investigates the effect of high concentrations of cellulose fibers, of non-negligible size, on the abrasion resistance performance of an acrylic paint.

Figure 12 shows the results of the scrub test carried out on the four coatings. A contained concentration of fibers (sample F5) involves an effective reduction in the amount of material removed from the coating compared to the pure acrylic matrix (sample F0). The hardness of this type of fiber, revealed in Figure 11, plays a key role in contrasting

abrasion phenomena. After 1000 scrub cycles, sample F5 exhibits a mass loss reduction of 41.8% compared to the filler-free coating. Sample F10, on the other hand, exhibits a slight increase in mass loss throughout the test. However, after 1000 scrub cycles, the result of coating F10 is comparable with that of layer F0. Instead, the outcome of sample F20 is completely different, as it reveals a very high loss of material after only 250 scrub cycles.

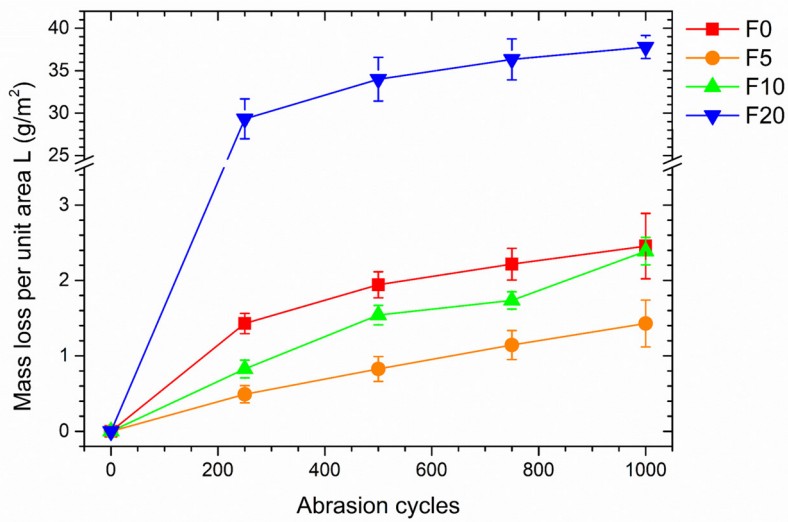

**Figure 12.** Loss in coatings mass per unit area, as a function of the number of abrasion cycles.

The behavior of the four samples is easily understood by observing Figure 13, which displays the SEM micrographs of their surfaces at the end of the test. Figure 13a (sample F0) reveals the typical traces of abrasion produced by the rubbing of the scrub test sponge. The surface appearance of sample F5 (Figure 13b) is very similar, but it shows the presence of some cellulose fibers. Despite the aggressive abrasion process, the excellent coherence between the polymeric matrix and the cellulose fibers allows the latter to remain well anchored to the coating. The hardness of the fibers allows them to resist the material removal process conducted by the scrub test sponge. This phenomenon explains the superior behavior of sample F5 observed in the graph in Figure 12. Otherwise, the surface of sample F10 (Figure 13c) shows some important defects. At the end of the test, the coating possesses large holes, which can even reach the wooden substrate. These holes represent the location sites of the large agglomerations of fibers shown earlier in Figure 2c. Since these accumulations represent significant dimensional asperities, they are easily removed by the continuous movement of the abrasive sponge. Consequently, the material loss evaluated during the scrub test is higher than that of the pure acrylic matrix. In this case, the protective properties of the fibers are countervailed by the defects they introduce into the coating. Finally, sample F20 (Figure 13d) reveals only residual traces of initial coating. A large part of the surface is in fact composed of the wooden substrate. The morphology of the coating completely distorted by the high amount of fibers means that the layer possesses poor adhesion and offers almost no resistance to the passage of the abrasive sponge. The inherent high porosity leads to the complete destruction of the coating and its removal during the test. The mass loss is high after only 250 cycles, as most of the coating has been broken down and removed by the sponge.

The cellulose fibers enable the increase in the hardness and abrasion resistance of organic coatings, as long as their concentration is kept below a maximum threshold. The two physical properties are interconnected: the increase in abrasion resistance is mainly due to the good intrinsic hardness of the fibers. However, both strongly depend on the structural morphology of the coating. High quantities of fibers upset the structure of the layer, introducing defects and porosity, which translate into poor resistance to the indenter and to the movement of the abrasive sponge. Consequently, it is not convenient to use a

quantity of fibers greater than 5 wt.%, in order to avoid altering the morphology of the coating and, consequently, its mechanical properties.

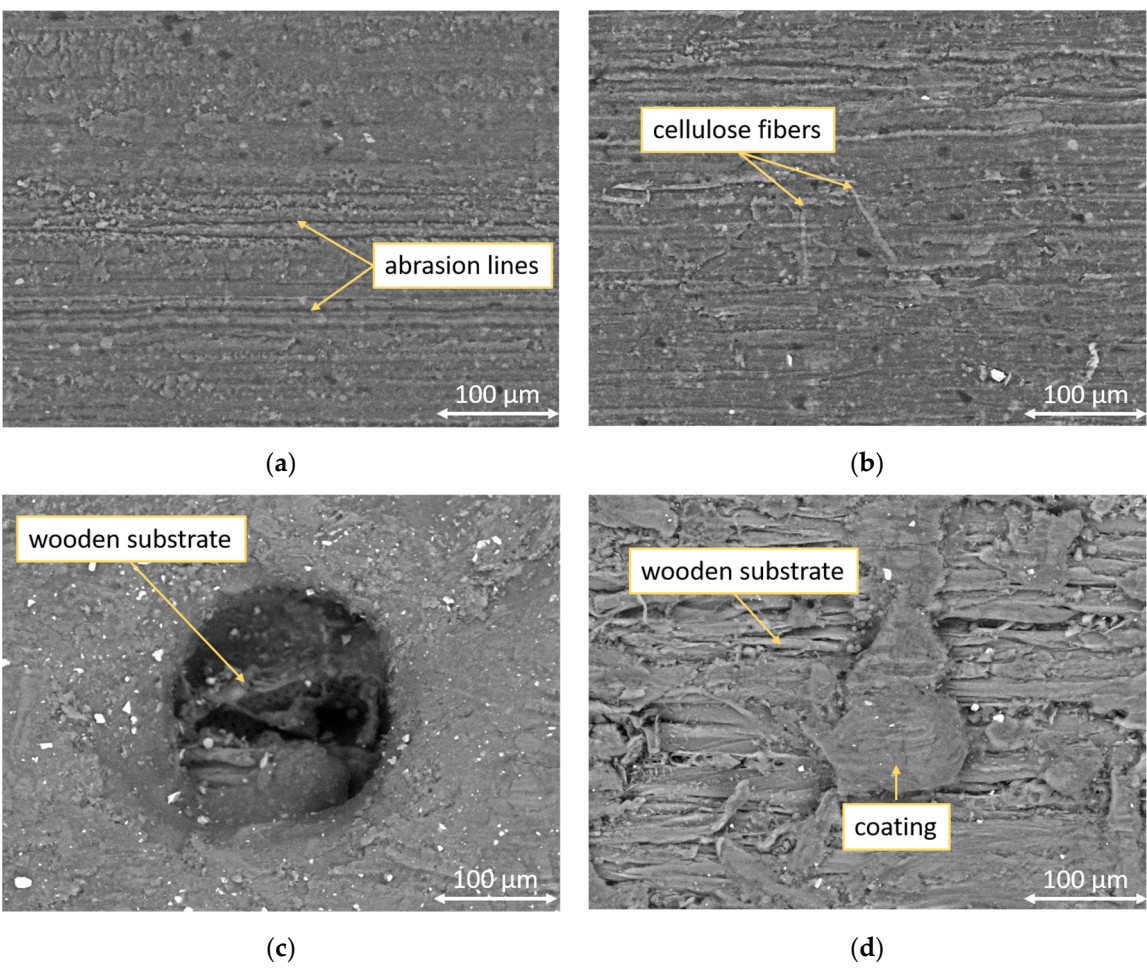

**Figure 13.** Top-view SEM micrographs of the surface morphology of (**a**) sample F0, (**b**) sample F5, (**c**) sample F10 and (**d**) sample F20 after the 1000 scrub test cycles.

## 4. Conclusions

This work reveals the consequences of employing high concentrations of cellulose fibers as fillers for wood paints. The bio-based filler involves a change in the rheology of the paint, with consequent distortion of the structure of the bulk coating. High amounts of fibers lead to the development of porosity in the layer, which does not appear compact and does not suggest adequate protective guarantees for the wooden substrate.

The coating undergoes a slight color change following exposure to UV-B radiation, whose penetration into the acrylic matrix is favored by the presence of the fibers. Moreover, the bio-based filler, especially in large quantities, partially inhibits the protective role of the acrylic matrix, due to their tendency to photo-oxidize as a result of their cellulose nature. Otherwise, the cellulose fibers do not influence the protective properties of the paint in environments with critical temperatures, as the four series of samples do not show phenomena of whitening or development of cracks during the test of exposure in the climatic chamber.

The liquid resistance test and the water uptake test manifest the influence of the fibers in reducing the barrier performance of the acrylic coating, as the filler introduces discontinuity in the polymeric matrix, favoring the uptake of liquids.

Finally, the cellulose fibers enable the increase in the hardness and abrasion resistance of organic coatings, as long as their concentration is kept below a maximum threshold. In fact, high quantities of fibers upset the structure of the layer, introducing defects and

porosity, which translates into a concrete reduction in the mechanical resistance of the composite coating.

In conclusion, this work disseminates the pros and cons of using large amounts of cellulose fibers. This material is particularly interesting as a filler for wood coatings because of its ecological nature and good mechanical characteristics, especially in this historical period when a new sustainable economy is being promoted. However, this work warns against the excessive use of these fibers, which need a threshold limit so as not to significantly change the coating's structure and thereby weaken its protective properties.

**Author Contributions:** Conceptualization, M.C. and S.R.; methodology, M.C. and S.R.; investigation, M.C.; data curation, M.C. and S.R.; writing—original draft preparation, M.C.; writing—review and editing, M.C. and S.R.; supervision, S.R. All authors have read and agreed to the published version of the manuscript.

**Funding:** This research received no external funding.

**Institutional Review Board Statement:** Not applicable.

**Informed Consent Statement:** Not applicable.

**Data Availability Statement:** The data presented in this study are available on request from the corresponding author. The data are not publicly available due to the absence of an institutional repository.

**Acknowledgments:** The authors greatly acknowledge the contributions of Stefano Di Blase (ICA Group, Civitanova Marche, Italy) and Stefano Beschi (Rettenmaier Italia, Castenedolo, Italy) regarding the paints and cellulose fibers supply, respectively. The publication was created with the co-financing of the European Union—FSE-REACT-EU, PON Research and Innovation 2014–2020 DM1062/2021.

**Conflicts of Interest:** The authors declare no conflict of interest.

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
