# Peer review of "Impact of High Concentrations of Cellulose Fibers on the Morphology, Durability and Protective Properties of Wood Paint"

_coatings, doi:10.3390/coatings13040721_

Round 1

Reviewer 1 Report

In figure 1 SEM pictures are shown. What is the substrate in this case? How was the film formed? The parameters in the black part of the pictures are way too little and cannot be read. This is true for all SEM pictures in the paper.

On page 5 in line 203 the concentration is given g/ml on the same page in line 204 the concentration is given in g/cm². Please use the same units throughout the manuscript.

In Figure 3 I think one should be able to e.g. compare F20 at 0h and F20 at 200h. In the current version of the figure this is not possible. Please regroup the spectra accordingly.

In the methods section a bit more description would be helpful. E.g. I have no idea what the cross cut test is, and the explanation currently given does not help.

Author Response

In figure 1 SEM pictures are shown. What is the substrate in this case? How was the film formed? The parameters in the black part of the pictures are way too little and cannot be read. This is true for all SEM pictures in the paper.

Authors: Figure 1 is representative of the cellulose fibers alone. The fibers were positioned on the appropriate SEM support, by gluing with conductive tape. Therefore, there is no film in the image, which is representative of the morphology of the fibers as they are. The authors added the sentence at line 183: “The fibers were positioned on the appropriate SEM support, by gluing with conductive tape”.

All the SEM images have been modified to better see the marker.

On page 5 in line 203 the concentration is given g/ml on the same page in line 204 the concentration is given in g/cm². Please use the same units throughout the manuscript.

Authors: the text has been corrected.

In Figure 3 I think one should be able to e.g. compare F20 at 0h and F20 at 200h. In the current version of the figure this is not possible. Please regroup the spectra accordingly.

Authors: the authors understand the reviewer's request, but such a distribution would make it difficult to compare different types of samples at the same time. In the current state of the graph, however, the same color of the curves allows an easy comparison between the behavior of the same sample at different times. After all, the graph demonstrates that there are no substantial changes in the spectra over time, while the comparison between the spectra of the four sets of painted samples is more interesting. So, the authors thank the reviewer for the advice, but have decided to keep the previous distribution. Rather, the authors have preferred to include the reference bands and peaks, to better understand the text.

In the methods section a bit more description would be helpful. E.g. I have no idea what the cross cut test is, and the explanation currently given does not help.

Authors: the authors thoroughly described the characterization techniques employed in the study. If they had to describe the functioning of the single test, the characterization section would become really too thick. For each test, the authors associated a norm that they followed. This usually allows the reproducibility of the measurements. However, the authors decided to satisfy the reviewer's requests by describing how the Cross Cut test works.

Reviewer 2 Report

The manuscript is dedicated to obtaining paints with a biodegradable filler - cellulose. This immediately raises the question of the durability of such paint? In the methodological section of the manuscript, it is important to indicate the composition of the wood used. Here, the content of resins and fats, lignin and hemicellulose is most interesting. It is very important that the authors note the effect of the filler on the rheological properties of the system, as rheology is one of the determining factors for paints.

A large amount of fiber introduced into the paint leads to its uneven distribution and the formation of defects.

For the samples obtained with different filler content, various natural effects are modeled, for example, hydrocarbons.

The main conclusion of the work is the limitation of the filler concentration in filled paints due to a critical change in the structure of the resulting coating and a decrease in the required properties.

In general, the work was done at a good level and can be published in a journal.

L.102. need to fix "filber"

Figures 1,2. The quality of the scale bar needs to be improved.

Figure 3. The presented spectra are of poor quality. I also recommend that the authors add the bands discussed in the text to the spectra, for the convenience of readers.

Figure 13. Why are some of the photographs showing the grain of the coating perpendicular to the grain of the wood, while others are parallel? It is also not clear why there are only two fibrils in photo b?

L.499, 529. "In conclusion" - I propose to delete.

Author Response

The manuscript is dedicated to obtaining paints with a biodegradable filler - cellulose. This immediately raises the question of the durability of such paint? In the methodological section of the manuscript, it is important to indicate the composition of the wood used. Here, the content of resins and fats, lignin and hemicellulose is most interesting. It is very important that the authors note the effect of the filler on the rheological properties of the system, as rheology is one of the determining factors for paints.

A large amount of fiber introduced into the paint leads to its uneven distribution and the formation of defects.

For the samples obtained with different filler content, various natural effects are modeled, for example, hydrocarbons.

The main conclusion of the work is the limitation of the filler concentration in filled paints due to a critical change in the structure of the resulting coating and a decrease in the required properties.

In general, the work was done at a good level and can be published in a journal.

L.102. need to fix "filber"

Authors: the authors corrected the word ‘fibers’.

Figures 1,2. The quality of the scale bar needs to be improved.

Authors: the images have been modified to better observe the scale bar.

Figure 3. The presented spectra are of poor quality. I also recommend that the authors add the bands discussed in the text to the spectra, for the convenience of readers.

Authors: the figure has been modified, as requested.

Figure 13. Why are some of the photographs showing the grain of the coating perpendicular to the grain of the wood, while others are parallel? It is also not clear why there are only two fibrils in photo b?

Authors: what does the reviewer mean by coating grain? The 4 images represent the surfaces of the samples observed in top-view. The images represent the horizontal lines of abrasion in the coating following the scrub test. In sample F5 it is still possible to observe the presence of some cellulose fibers in the coating (indicated by the arrows). Otherwise, sample F10 shows real holes in the coating, where the underlying wooden substrate can be observed. Finally, sample F20 has lost most of its coating, which can only be observed in limited areas. The images and the relative caption have been slightly modified to better highlight what is observed and to specify that the images refer to the top-view of the samples.

L.499, 529. "In conclusion" - I propose to delete.

Authors: the authors deleted the words at line 499, but they prefer to keep them at line 529, as useful for summarizing the conclusions of the study.
